# Lung IL-17A-Producing CD4^+^ T Cells Correlate with Protection after Intrapulmonary Vaccination with Differentially Adjuvanted Tuberculosis Vaccines

**DOI:** 10.3390/vaccines12020128

**Published:** 2024-01-26

**Authors:** Erica L. Stewart, Claudio Counoupas, Diana H. Quan, Trixie Wang, Nikolai Petrovsky, Warwick J. Britton, James A. Triccas

**Affiliations:** 1Sydney Infectious Diseases Institute (Sydney ID), Faculty of Medicine and Health, The University of Sydney, Camperdown, NSW 2006, Australia; e.stewart@centenary.org.au (E.L.S.); claudio.counoupas@sydney.edu.au (C.C.); 2Centre for Infection and Immunity, Centenary Institute, Royal Prince Alfred Hospital, Camperdown, NSW 2050, Australia; d.quan@centenary.org.au (D.H.Q.); t.wang@centenary.org.au (T.W.); w.britton@centenary.org.au (W.J.B.); 3School of Medical Sciences, Faculty of Medicine and Health, The University of Sydney, Camperdown, NSW 2006, Australia; 4Centre for Inflammation, School of Life Sciences, Faculty of Science, The University of Technology Sydney, Ultimo, NSW 2007, Australia; 5Vaxine Pty Ltd., Warradale, Adelaide, SA 5046, Australia; nikolai.petrovsky@vaxine.net; 6Department of Clinical Immunology, Royal Prince Alfred Hospital, Camperdown, NSW 2050, Australia

**Keywords:** tuberculosis, vaccine, adjuvant, mucosal, Th17, IL-17A, correlates

## Abstract

Tuberculosis (TB), caused by *Mycobacterium tuberculosis*, results in approximately 1.6 million deaths annually. BCG is the only TB vaccine currently in use and offers only variable protection; however, the development of more effective vaccines is hindered by a lack of defined correlates of protection (CoP) against *M. tuberculosis*. Pulmonary vaccine delivery is a promising strategy since it may promote lung-resident immune memory that can respond rapidly to respiratory infection. In this study, CysVac2, a subunit protein previously shown to be protective against *M. tuberculosis* in mouse models, was combined with either Advax^®^ adjuvant or a mixture of alum plus MPLA and administered intratracheally into mice. Peripheral immune responses were tracked longitudinally, and lung-local immune responses were measured after challenge. Both readouts were then correlated with protection after *M. tuberculosis* infection. Although considered essential for the control of mycobacteria, induction of IFN-γ-expressing CD4^+^ T cells in the blood or lungs did not correlate with protection. Instead, CD4^+^ T cells in the lungs expressing IL-17A correlated with reduced bacterial burden. This study identified pulmonary IL-17A-expressing CD4^+^ T cells as a CoP against *M. tuberculosis* and suggests that mucosal immune profiles should be explored for novel CoP.

## 1. Introduction

Tuberculosis (TB) remains a major global health burden, and the currently used vaccine, *Mycobacterium bovis* bacille Calmette-Guerin (BCG), provides only variable protection [1,2]. There are multiple vaccine strategies currently being pursued for the clinical development of new TB vaccines, but the best efficacy achieved thus far in a human clinical trial has been 49.7% protection against active pulmonary TB by the subunit vaccine candidate M72/AS01E [3]. Many challenges complicate the development of new vaccines. For example, protection against infection requires years-long, expensive clinical trials in high-disease-burden settings. Furthermore, most adults in these regions are not immunologically naïve (due to prior BCG immunisation and/or exposure to *M. tuberculosis*), meaning novel vaccines must be effective as booster immunisations [2]. However, a major roadblock to the development of effective new vaccines is a lack of correlates of protection (CoP) against *M. tuberculosis* infection. It is well established that IFN-γ is essential for the control of mycobacterial infections, and primary immunodeficiency in the IFN-γ, IL-12 or STAT-1 pathways results in increased susceptibility to infection with mycobacteria or *Salmonella* [4]. Although the presence of multifunctional Th1-polarised CD4^+^ T cells co-expressing IFN-γ, TNF and IL-2 has been considered a CoP against *M. tuberculosis* infection, vaccines inducing these responses do not consistently protect against infection in human trials [5,6,7,8]. Thus, new CoP are required for the development of effective vaccines against TB.

Whilst parenteral vaccine administration generates an adequate immune response against most pathogens, thus far it has been insufficient to fully protect against *M. tuberculosis*. Traditional vaccine delivery methods that target systemic immunity, such as parenteral delivery of BCG, generate robust circulating antibody responses and cellular immune memory at the vaccine site-draining lymph node (LN). This strategy is sufficient in most cases to protect against the miliary disseminated form of TB, but, since *M. tuberculosis* can delay the response time for T cells to translocate from the LN to the lungs, BCG is often unable to prevent pulmonary TB [9,10,11]. Thus, a strategy of interest is the delivery of vaccines directly to the site of the infection, via either intranasal or intrapulmonary (i.e., intratracheal [IT]) administration. In particular, the induction of tissue-resident memory T cells (TRMs) that can respond rapidly to inhaled pathogens is a primary goal of mucosal vaccination and has been shown to be advantageous in protection against *M. tuberculosis* in animal models [8,11,12]. Recent studies in rhesus macaques examining intravenous BCG administration revealed that the generation of pulmonary antigen-specific CD4^+^ T cells was associated with protective efficacy [13,14]. The first clinical trial of an adenoviral-vectored *M. tuberculosis* vaccine candidate administered via the aerosol route showed encouraging results for the translation of pulmonary vaccines, primarily the induction of polyfunctional CD4^+^ TRMs in the airways [15]. Therefore, mucosal delivery is a promising strategy to generate effective immunity against *M. tuberculosis*.

Despite the evidence for the generation of TRMs as a promising CoP for *M. tuberculosis* vaccines, the current TB vaccine pipeline primarily consists of parenterally administered vaccines that promote a systemic Th1 response [16,17]. Thus, there is a need for more diverse vaccine candidates to enter the clinical pipeline. In this study, a major aim was to characterise the protective efficacy of a subunit TB vaccine combined with two different adjuvants: a combination of alum and MPLA that models the adjuvant AS04 used in the hepatitis B vaccine Fendrix^®^, and Advax^®^, a polysaccharide adjuvant based on delta inulin and studied in multiple vaccine clinical trials [18,19,20,21]. Both vaccines contained the fusion protein CysVac2, consisting of *M. tuberculosis* Ag85B and CysD, which induces protection against *M. tuberculosis* challenge in animal models when administered parenterally and to the mucosa [11,19,22]. In this study, intrapulmonary CysVac2 vaccines using different adjuvants promoted lung IL-17A-producing CD4^+^ T cells with a similar phenotype after *M. tuberculosis* challenge. In addition, the presence of pulmonary Th17 correlated with protection against *M. tuberculosis* infection, while systemic or lung-local IFN-γ-expressing CD4^+^ T cells did not. Hence, these data identified pulmonary CD4^+^ T cells expressing IL-17A as a CoP for *M. tuberculosis* infection and support the progression of aerosol-delivered CysVac2 vaccines to clinical trials.

## 2. Materials and Methods

### 2.1. Mice and Immunisations

Female C57BL/6 mice 6–8 weeks of age were sourced from Animal Resources Centre (Perth, WA, Australia) and were maintained in specific pathogen-free conditions at the Centenary Institute (Sydney, NSW, Australia). For all experiments, mice were selected at random for experimental groups and were administered treatment in random order. All mouse experiments were approved by the Sydney Local Health District Animal Ethics and Welfare Committee (protocol 2020-009). Advax particles were provided by Vaxine Pty Ltd. (Adelaide, South Australia, Australia) at 50 mg/mL. Alum/MPLA-adjuvanted vaccines were prepared by mixing alum in the form of Alhydrogel^®^ adjuvant 2% (InvivoGen, San Diego, CA, USA) and monophosphoryl lipid A from *S. minnesota* R595 (InvivoGen). CysVac2 protein was recombinantly expressed in *E. coli* by The University of Sydney Analytical Core Facility (Sydney, NSW, Australia). For IT vaccinations, mice were anaesthetised with an intraperitoneal (IP) injection with ketamine (80 mg/kg) and xylazine (10 mg/kg) in PBS followed by IT instillation with 10 μg CysVac2 antigen mixed with 1 mg Advax, with 20 μg Alhydrogel^®^ adjuvant 2% and 2 μg MPLA, or with 5 μg Alhydrogel^®^ adjuvant 2% and 0.5 μg MPLA in a total volume of 50 μL in endotoxin-free PBS (Sigma-Aldrich) using the PennCentury Microsprayer Aerosoliser (PennCentury, Wyndmoor, PA, USA). In all experiments, mice vaccinated with endotoxin-free PBS (Sigma-Aldrich, St. Louis, MO, USA) alone served as controls.

### 2.2. Blood Sample Collection and Analysis

Blood samples were collected from the tail vein into tubes containing 10 μL heparin (50 U, Sigma-Aldrich), centrifuged to separate plasma, and peripheral blood mononuclear cells (PBMCs) isolated using Histopaque 1083 (Sigma-Aldrich). For antigen recall, PBMCs were restimulated with a final concentration of 5 μg/mL CysVac2 in supplemented RPMI 1640 medium, HEPES (Life Technologies, Thermo Fisher Scientific, Carlsbad, CA, USA). Cells were incubated for 4 h at 37 °C before the addition of Protein Transport Inhibitor Cocktail (Life Technologies, Thermo Fisher Scientific) and then incubation overnight at 37 °C. Restimulated PBMCs were then stained for intracellular cytokine production. Alternatively, unstimulated PBMCs were stained intracellularly immediately after collection for transcription factor expression.

### 2.3. Preparation of Tissue Samples for Flow Cytometry

Lungs were perfused with chilled PBS and then added to RPMI media (Life Technologies, Thermo Fisher Scientific) containing 10 U/mL DNAse I and Collagenase IV (Sigma-Aldrich). Lungs were then mechanically dissociated using a GentleMACS dissociator (Miltenyi Biotec, Sydney, NSW, Australia), before being incubated for 30 min at 37 °C. After incubation, lungs were further dissociated through a 70 μm nylon cell strainer, washed and then resuspended in supplemented RPMI. Red blood cells were removed from single-cell suspension through the addition of 1 mL ACK lysis buffer (Thermo Fisher Scientific), incubation at room temperature (RT) for 45 s and quenching of the reaction with RPMI with 5% FCS. Cells were then washed again with RPMI and resuspended in FACS buffer prior to antibody staining. LNs were prepared with DNAse I and Collagenase IV digestion (10 U/mL, Sigma-Aldrich) and incubation for 20 min at 37 °C. Following incubation, LNs were dissociated through a 70 μm nylon sieve, washed with RPMI and resuspended in FACS buffer prior to antibody staining. Samples were spiked with a known number of Rainbow Calibration Beads (Becton Dickinson Macquarie Park, NSW, Australia), filtered, then run on an LSRII 5L cytometer (BD Biosciences).

### 2.4. Staining of Cells for Flow Cytometry

Cell suspensions were first stained for cell surface markers using monoclonal antibodies (mAbs) (detailed in Appendix A), Fixable Blue Dead Cell stain (Life Technologies, Thermo Fisher Scientific) and anti-CD16/32 blocking antibody (clone 2.4G2; Becton Dickinson) diluted in FACS buffer for 30 min on a shaker at 4 °C. Samples were washed three times with FACS buffer and resuspended in 10% neutral buffered formalin (NBF; Sigma-Aldrich) if only surface stained and fixed for at least 1 h. Samples to be stained intracellularly for cytokine expression were first permeabilised using the BD Cytofix/Cytoperm Fixation/Permeabilization Kit (Becton Dickinson) as per the manufacturer’s instructions. This entailed incubating the samples in permeabilisation solution for twenty minutes at 4 °C, followed by washing twice in Perm/Wash buffer (Becton Dickinson). Samples were stained with a cocktail of mAbs specific for various cytokines for 45 min at 4 °C on a shaker. Cells were then washed and resuspended in 10% NBF (Sigma-Aldrich) prior to flow cytometric analysis. For analysis of transcription factor expression, after surface staining, cells were fixed and permeabilised using the eBioscience^TM^ Foxp3/Transcription Factor Staining Buffer Set (Invitrogen, Thermo Fisher Scientific). Following fixation and permeabilisation, cells were incubated with a mix of mAbs for transcription factors for 45 min on a shaker at RT. Cells were then washed and resuspended in 10% NBF (Sigma-Aldrich) for flow cytometric analysis. For any samples that were exposed to *M. tuberculosis*, after removal from the PC3 facility, additional 10% NBF was added to the samples prior to analysis on the flow cytometer. All flow cytometry was performed on an LSRII 5L cytometer (BD Biosciences, Franklin Lakes, NJ, USA).

### 2.5. Flow Cytometric Data Analysis

Flow cytometric data were analysed using FlowJo Software version 10.9.0 (Becton Dickinson). Manual gating strategies are depicted in Appendix A. For UMAP analysis of myeloid cells in the lungs, 20,000 events per sample were concatenated and clustered based on the expression of Ly6C, CD11c, B220, CD11b, Ly6G, CD103, Siglec-F and CD64 using FlowJo Software (Becton Dickinson). UMAP gating was validated with the manual gating strategy depicted in Appendix A. For some experiments, FlowJo Software was also used to perform Boolean gating on samples to determine co-expression of cytokines. Flow cytometric data were normalised across experiments using the formula $z=\frac{\left(original\;value-mean\;of\;biological\;replicates\right)}{\left(standard\;deviation\;of\;experiment\right)}$ [23] (Jaadi, 2021). These data were then input into the web tool ClustVis [24], which generates a heatmap and PCA plot.

### 2.6. Antibody Enzyme-Linked Immunosorbent Assays (ELISAs)

Plasma samples were analysed for anti-CysVac2 antibody titres by coating Corning 96-Well Clear PVC Assay Microplates (Sigma-Aldrich) with 5 µg/mL CysVac2 in PBS overnight at RT. The next day, ELISAs were performed as previously described [25]. Titres were determined using GraphPad Prism 9 software (GraphPad Software Inc., Boston, MA, USA) to fit a sigmoidal curve and calculate the intersection with three standard deviations above the mean negative control value (average absorbances of unvaccinated mouse plasma).

### 2.7. Statistical Analysis

Statistical analysis was performed using GraphPad Prism^®^ version 10 (GraphPad Software Inc.). For multi-group datasets, 1-way ANOVA was used, and 2-way ANOVA was used for time-course experimental data. For 1-way ANOVA, the Tukey post-hoc test was used to correct for multiple comparisons, and for 2-way ANOVA, the Dunnett, Holm–Sidak’s or Sidak’s multiple comparisons tests were used where appropriate. Correlates analysis was also performed using GraphPad Prism software. Data were analysed using a Spearman’s correlation test or correlation matrix, where *p* < 0.05 was considered significant.

## 3. Results

### 3.1. Intratracheal Vaccination with Differently Adjuvanted CysVac2 Vaccines Induces Similar Circulating Immune Responses

The antigen-specific cytokine production of PBMCs is indicative of the memory response being generated and is often used to measure immunogenicity in clinical trials of TB vaccine candidates [16]. Mice were vaccinated three times, 2 weeks apart, and then rested for 4 weeks before being challenged with aerosol *M. tuberculosis* (Figure 1A). The vaccines used in this study were CysVac2 (10 µg) with a low dose of alum and MPLA (5 µg alum mixed with 0.5 µg MPLA, denoted CysVac2/AlumMPLA_low_), CysVac2 with a high dose of alum and MPLA (20 µg alum mixed with 2 µg MPLA, denoted CysVac2/AlumMPLA_high_) and CysVac2 with 1 mg of Advax (CysVac2/Advax). Some animals were immunised with PBS as a control, and all experimental mice were immunised via the IT route. At time points after each immunisation outlined in Figure 1A, mice were bled and plasma and PBMCs were separated. PBMCs were then restimulated with CysVac2 protein to determine antigen-specific cytokine responses measuring IFN-γ, IL-2, TNF, IL-17A and IL-10 in both CD4^+^ and CD8^+^ T cells using the gating strategy depicted in Appendix A. As shown in Figure 1B–E, vaccination with CysVac2/Advax and CysVac2/AlumMPLA_high_ induced similar kinetics of cytokine expression across the vaccination schedule. CD4^+^ IFN-γ expression for both vaccines peaked after early vaccinations (either the first or second) (Figure 1B). CD4^+^ IL-2 and TNF expression were similar between the CysVac2/Advax and CysVac2/AlumMPLA_high_ groups, both peaking after the second vaccination (Figure 1C,D). The kinetics of CD4^+^ IL-17A expression differed between the two vaccines somewhat, with levels in CysVac2/AlumMPLA_high_ mice remaining steady throughout the immunisation schedule and CysVac2/Advax showing a more delayed response, peaking after the third vaccination (Figure 1E). CysVac2/AlumMPLA_low_ vaccination showed a reduced peak in cytokine expression compared to CysVac2/AlumMPLA_high_ and CysVac2/Advax (Figure 1B–D), except for IL-17A expression, which peaked after the second immunisation (Figure 1E). For all vaccines, antigen-specific CD4^+^ IL-10 expression was not detected at any time point, nor was CD8^+^ T cell cytokine expression (Appendix A). Thus, PBMC cytokine expression at time points after IT immunisation reveals relatively similar kinetics of circulating CD4^+^ T cell functionality induced by two differently adjuvanted vaccines.

Recent studies have suggested that antibodies may play a role in protection against TB and thus, may be important to measure as part of vaccine development studies [13,26]. Only CysVac2/AlumMPLA_high_ generated significantly higher levels of plasma IgA one week after the final booster vaccination (Figure 1F). IgG1 and IgG2c antibodies, indicative of Th2 and Th1 responses, respectively, were also measured in the plasma at time points after vaccination (Figure 1G–I). There were no significant differences among the three groups; however, CysVac2/Advax and CysVac2/AlumMPLA_high_ showed a trending increased titre compared to CysVac2/AlumMPLA_low_ (Figure 1G,H). All vaccine groups had greater IgG1 titres than IgG2c (Figure 1I), suggesting a more Th2-polarised response than Th1. For all vaccines, antibodies were also detectable up to 7 weeks after the first immunisation, indicating a lasting systemic humoral response. Thus, all the vaccines tested in this study generated a robust antigen-specific IgG1 antibody response and a lesser IgG2c response that was detectable at late time points after immunisation.

### 3.2. CysVac2/Advax and CysVac2/AlumMPLA Are Protective in the Lungs When Administered Intratracheally, and Protection Is Associated with IL-17A Expression by Pulmonary CD4^+^ T Cells

Since vaccine-induced lung-resident TRMs have been previously identified to corelate with protection against *M. tuberculosis* infection [11,12], this study aimed to investigate if they are generated after IT vaccination with diverse adjuvants. Four weeks after the final vaccination, mice were challenged with ~100 CFU *M. tuberculosis* H37Rv via the aerosol route (as outlined in Figure 1A) and their immunity was examined 4 weeks post-challenge. IT immunisation with CysVac2/AlumMPLA_high_ and CysVac2/Advax significantly reduced the bacterial load in the lungs compared to PBS-immunised controls, while CysVac2/AlumMPLA_low_ did not (Figure 2A). None of the vaccines tested provided protection in the spleen (Figure 2B).

To determine the functionality of CD4^+^ T cells in the lungs after challenge, lung single-cell suspensions were restimulated with CysVac2 protein. Intracellular cytokine staining was performed for IFN-γ, IL-17A, TNF and IL-2 expression after restimulation, and Boolean gating was used to identify cells co-expressing combinations of multiple cytokines (Figure 2C). It was observed that in PBS control mice infected with *M. tuberculosis*, the majority of CD4^+^ T cells responding to CysVac2 restimulation expressed IFN-γ, either alone or in combination with other cytokines, primarily TNF and/or IL-2 (Figure 2C,E,I, J). In contrast, a distinct shift towards IL-17A expression was observed in the CD4^+^ T cells responding to CysVac2 from the mice of vaccinated groups (Figure 2C,F,L). The majority of CD4^+^ T cells producing IL-17A expressed it alone or in combination with TNF; however, there was also a subset CD4^+^ T cells expressing IL-17A alongside TNF and IL-2 (Figure 2C,K). The pattern of CD4^+^ T cell cytokine expression in all vaccinated groups was highly similar and in distinct contrast to unvaccinated PBS controls (Figure 2C). In CysVac2/AlumMPLA_high_ and CysVac2/Advax-immunised mice, the capacity for CD4^+^ T cells to express any of the cytokines measured was also enhanced compared to both PBS and CysVac2/AlumMPLA_low_-vaccinated groups (Figure 2D). Furthermore, the proportion of CD4^+^ T cells expressing IL-17A (either alone or in combination with other cytokines) was enhanced in all vaccinated groups, particularly the CysVac2/AlumMPLA_high_ and CysVac2/Advax groups (Figure 2C). Conversely, the presence of multifunctional CD4^+^ T cells expressing IFN-γ, TNF and IL-2 was reduced in all vaccinated mice compared to PBS-immunised control mice (Figure 2J). Instead, there was a significant increase in CD4^+^ T cells expressing IL-17A, TNF and IL-2 concurrently (Figure 2K). In the lung CD8^+^ T cell compartment, the only experimental group that showed any change in cytokine expression compared to PBS control mice was IT CysVac2/Advax-immunised animals (Appendix A). In these animals, a significantly greater expression of IFN-γ and IL-17A in lung CD8^+^ T cells was observed (Appendix A). In total, IT vaccination with CysVac2 vaccines caused a distinct shift in pulmonary CD4^+^ multifunctionality from cells expressing IFN-γ towards IL-17A-producing cells after *M. tuberculosis* challenge, irrespective of the adjuvant used.

RorγT is the master regulator transcription factor controlling Th17 and Tc17 cell differentiation, while T-bet is the master regulator for Th1 and Tc1 differentiation [27]. *M. tuberculosis* infection is known to induce a Th1 response in the lung, and this was confirmed in all the groups, with the PBS control group showing the highest percentage of T-bet expression in both CD4^+^ and CD8^+^ T cells (Figure 3A,C). Conversely, in the vaccinated groups, robust expression of RorγT in CD4^+^ T cells was observed in the lungs, coupled with a downregulation of T-bet expression compared to PBS control mice (Figure 3A,B). Interestingly, only CysVac2/Advax induced the upregulation of RorγT in lung CD8^+^ T cells, which was not seen in any other group (Figure 3D). In these pulmonary CD8^+^ T cells, there was a similar downregulation of T-bet in all vaccinated groups as was observed in CD4^+^ T cells (Figure 3C). T cells in the lungs were also measured for TRM-like marker expression (CD44^+^CD69^+^CD62L^−^ for CD4^+^ T cells and CD103^+^CD44^+^CD69^+^CD62L^−^ for CD8^+^ T cells) (Appendix A). There were no significant changes compared to PBS control mice in the proportions of T cells with TRM-like markers in the lungs at this time point. Further, there were no significant changes in the proportion of B cells, germinal centre B cells or memory B cells in the lungs at this time point (Appendix A). However, the CysVac2/Advax-immunised mice showed a significant increase in class-switched (IgM^−^IgD^−^) B cells in the lungs (Appendix A). Restimulation of single-cell suspensions from the mLN showed that a similar Th17 response was induced in IT-vaccinated mice, defined by CD4^+^ T cell IL-17A and TNF expression (Figure 4A) and expression of RorγT (Figure 4B). Unlike the lungs, there was no significant decrease in the expression of T-bet in mLN CD4^+^ T cells at this time point (Figure 4C). Thus, IT vaccination with all vaccines induced a distinct upregulation of RorγT in the lungs and mLN, with downregulation of T-bet in lung CD4^+^ T cells after challenge. However, CysVac2/Advax additionally promoted the expression of RorγT and IL-17A in pulmonary CD8^+^ T cells and the induction of local germinal centre B cells.

### 3.3. Intratracheal Vaccination with CysVac2/Advax and CysVac2/AlumMPLA Generates Similar Myeloid Cell Recruitment to the Lungs after M. tuberculosis Challenge

Infiltrating cells such as neutrophils, monocytes and eosinophils are recruited to the lung after *M. tuberculosis* infection and have important roles in protection or immunopathology [28,29]. A multicolour flow cytometry panel was used to identify numerous myeloid cell subsets using the gating strategy in Appendix A. To visualise all the populations identified concurrently, the nonlinear dimensionality-reduction technique known as uniform manifold approximation and projection (UMAP) was used to cluster the cells based on multiple phenotypic markers (Figure 5A). The populations identified in Figure 5A were verified using manual gating techniques. Like many of the T cell parameters observed, the myeloid profile of the lungs 4 weeks after challenge was highly similar between immunised animals irrespective of the vaccine administered (Figure 5B–I). There was no significant difference in the recruitment of alveolar macrophages, eosinophils, neutrophils, cDC1s or MHCII-monocytes at this time point compared to unvaccinated control mice (Figure 5B–D,F,I). However, there was a significant reduction in certain monocytic and DC subsets in the lungs of vaccinated mice, including interstitial CD64^+^ macrophages, cDC2s and Ly6C^hi^ monocytes (Figure 5E,G,H).

The lung-draining mediastinal lymph nodes (mLN) were also examined for myeloid cell recruitment; however, there were no significant changes compared to PBS controls observed in the IT immunised mice (Appendix A). Hence, at this time point, there was not any significant recruitment of myeloid subsets associated with vaccine protective efficacy.

### 3.4. Intratracheal CysVac2/Advax and CysVac2/AlumMPLA_high_ Generate Distinct Immunological Profiles after M. tuberculosis Challenge, but a Shared Th17 Signature

To define in detail the immune signatures of the lungs post-*M. tuberculosis* infection, the flow cytometric data from independent experiments described above were standardised using the formula $z=\frac{\left(original\;value-mean\;of\;biological\;replicates\right)}{\left(standard\;deviation\;of\;experiment\right)}$ [23]. The data were then inputted into the web tool ClustVis [24], which generates a heatmap and principle component analysis (PCA) plot. To understand the signatures associated with protective responses compared to non-protective responses, the readouts of CysVac2/AlumMPLA_high_- and CysVac2/Advax- versus PBS-vaccinated mice were compared (Figure 6). In the PCA plot, the PBS control mice clustered separately from CysVac2/AlumMPLA_high_- and CysVac2/Advax-immunised mice, which clustered closely (Figure 6A). The lungs of PBS-immunised mice showed enriched cellular infiltration, including monocytes and cDC2 cells (Figure 6B). Vaccinated animals, conversely, showed enhanced adaptive immune responses but particularly the presence of RorγT^+^ and IL-17A expressing CD4^+^ T cells, which were noticeably reduced in PBS control mice (Figure 6B). Interestingly, signatures associated with macrophage activation, such as increased T-bet^+^ CD4^+^ T cells, CD4^+^ IFN-γ expression and CD64^+^ interstitial macrophages, were more enriched in PBS control mice compared to IT immunised animals. Other readouts, such as B cell phenotypes and CD8^+^ T cells, did not cluster distinctly with vaccination status and lung CFU. Overall, the signature associated with reduced lung bacterial burden was a Th17, CD4^+^ T cell-enriched phenotype.

### 3.5. Pulmonary CD4^+^ T Cells Expressing IL-17A Correlate with Protection against M. tuberculosis but IFN-γ-Expressing CD4^+^ T Cells Do Not

To further define CoP associated with effective TB vaccines, antigen-specific T cell functionality was examined in PBMCs and was correlated with lung bacterial burden after challenge using a Spearman’s correlation test, as described previously [30]. Shown in Figure 7A–D are the data from the peak of the circulating T cell response in week 3 (1 week post the first boost). In the blood, none of the cytokines measured expressed by CD4^+^ T cells significantly correlated with protection; however, IL-17A expression approached significance with a *p* value of 0.0519 (Figure 7A–D). CD4^+^ T cell responses in the lungs after infection were also analysed using Spearman’s correlation analysis with lung bacterial burden 4 weeks after challenge (Figure 7E). 

These data showed that the presence of CD4^+^ T cells in the lungs negatively correlated with lung bacterial burden, especially IL-17A-expressing CD4^+^ T cells expressing IL-2, RorγT and IL-17A (Figure 7E). Additionally, the presence of CD8^+^ T cells expressing RorγT and CD19^+^ B cells also correlated with protection. When examining Boolean gating of CD4^+^ T cell cytokine expression, the overall ability of CD4^+^ T cells to express cytokines also significantly correlated with reduced bacterial burden, as did polyfunctional cells co-expressing IL-17A, TNF and IL-2 (Figure 7F,H). Conversely, the presence of CD4^+^ T cells expressing IFN-γ or co-expressing IFN-γ, TNF and IL-2 did not correlate with reduced CFU in the lungs, and CD64^+^ macrophages correlated with increased bacterial burden (Figure 7E,G). Thus, these data indicate that CoP against *M. tuberculosis* may be more reliable when measured in the lungs compared to the blood and that pulmonary polyfunctional IL-17A-expressing CD4^+^ T cells are a potential CoP for *M. tuberculosis.*

## 4. Discussion

The peripheral antigen-specific T cell cytokine response is routinely analysed as a measure of immunogenicity in the assessment of TB vaccine candidates. A comparison of six TB vaccines that had advanced to clinical trials showed that they all induced primarily a Th1-polarised response; thus, there is a need for more diversity in the clinical pipeline [16]. In this study, IT immunisation with CysVac2/Advax and CysVac2/AlumMPLA_high_ both induced CD4^+^ T cells expressing IL-2, TNF and/or IL-17A (Figure 1 and Figure 2). Th17 cells are generated in the presence of IL-6, TGF-β and IL-23 [31], and it appears that mucosal immunisation favours Th17 cell polarisation even for vaccines that promote Th1 responses after parenteral administration [32]. In the current study, all the vaccines tested also promoted higher levels of CysVac2-specific IgG1 than IgG2c in the plasma of vaccinated animals (Figure 1). This phenotype after intrapulmonary vaccination has been observed in previous studies [33], and the combined expression of IL-17A and IL-21 (both expressed by Th17 cells) has been shown to preferentially promote IgG1 class switching [34]. Also associated with pulmonary vaccine delivery is the induction of IgA, the antibody class most found at mucosal sites, including the lungs [33]. In the current study, only modest IgA responses were observed in week 5 of the experiment (after the second boost), and this was only in the CysVac2/AlumMPLA_high_ mice. In contrast, previous investigations of Advax in IT administered vaccines have shown the induction of IgA both in the respiratory tract and in the circulation [11,35]. It is possible that some CysVac2-specific IgA was generated in the respiratory mucosa that was unable to be detected in the blood. Thus, both AlumMPLA and Advax promote circulating Th17 responses and IgG1-skewed humoral immune responses; the exact mechanism by which this occurs is of interest for future studies.

Historically, the generation of Th1 cells has been a focus of TB vaccine development, owing to their ability to secrete IFN-γ, activating macrophages for enhanced microbial killing. BCG, the currently used tuberculosis vaccine, induces significant levels of IFN-γ-expressing CD4^+^ T cells when administered intradermally [36]. Polyfunctional CD4^+^ T cells co-expressing IL-2, TNF and IFN-γ, like those generated by BCG, were historically considered a key CoP against *M. tuberculosis*; however, in an efficacy clinical trial, parenteral MVA85A induced multifunctional CD4^+^ T cells but did not protect against disease [7]. In the current study, we did not include a comparison with BCG. This is because we have previously shown that IT CysVac2/Advax generates equivalent protection in the lungs compared to subcutaneous BCG in C57BL/6 mice [11]. Furthermore, given that the mechanisms of protection induced by mucosal and parenteral vaccination are thought to be substantially different, we did not consider subcutaneous BCG an appropriate control in the current study. Other studies have examined pulmonary BCG delivery and found that lung IL-17A is a central mediator of protection [30,37,38]. Similarly, in the current study, it was observed that induction of Th1 cells did not correlate with protection, and in fact, a reduced Th1 phenotype was observed in the protected animals (Figure 2 and Figure 3). This was replaced with a robust Th17 signature, with the cytokine expression profiles of all vaccinated groups almost identical irrespective of the adjuvant used (Figure 2). The route of delivery appears to be a major contributing factor to the observed immune signature induced by CysVac2/Advax and other TB vaccines administered mucosally [25,32,37,39]. Previously, we have shown that intramuscular delivery of CysVac2/Advax generates multifunctional CD4^+^ T cells expressing IFN-γ, TNF and IL-2 but IT administration instead promotes a lung-local Th17 signature [11,19]. It would be of interest to examine if the immune signature we have identified here correlates with protective efficacy after parenteral vaccination, or if it is specific to pulmonary vaccine models.

In this study, IFN-γ expression by CD4^+^ T cells from the blood or in the lungs after challenge did not correlate with reduced lung bacterial burden (Figure 7). Recent studies in mice have indicated that excessive Th1 polarisation may result in overly differentiated T cells unable to efficiently migrate to the parenchyma of the lungs, thus hindering effective immune memory responses [40,41]. Th17 cells, conversely, can become TRMs in the lungs and enhance the early recruitment of Th1 cells into the tissue [33,42]. In this study, robust expression of the master regulator of Th17 cells, RorγT, was observed in lung CD4^+^ T cells in all IT immunised mice (Figure 3). Additionally, distinct RorγT expression was present in a subset of CD8^+^ T cells in the lungs of CysVac2/Advax-immunised animals after challenge, which was not present in any other study group (Figure 3). These cells could be Tc17 cells or MAIT cells; this experiment did not include the required markers to distinguish the two cell types, and the cytokine expression patterns of Tc17 and MAIT cells are very similar [43,44].

A notable finding from immune phenotyping was the relative enrichment of monocytes, cDC2 cells and CD64^+^ macrophages in the lungs of PBS control mice that were challenged with *M. tuberculosis* compared to vaccinated animals (Figure 5). While some monocytes can destroy *M. tuberculosis*, others are permissive to infection and may act as a niche for bacterial replication [45]. Thus, it is possible that high *M. tuberculosis* bacterial loads may recruit macrophages to the lungs that are less capable of controlling the infection; one study found that circulating monocytes isolated from TB patients were less capable of differentiating into DCs and stimulating T cell responses [46]. Another study observed that *M. tuberculosis* infection was able to inhibit antigen presentation to CD4^+^ T cells in myeloid DCs, despite high expression of MHC-II and costimulatory factors in these cells [47]. Given the broad heterogeneity of monocytes and myeloid DCs during infection and inflammation, it would be of interest in future studies to explore the functional capacity of the monocytes recruited to the lungs of vaccinated versus unvaccinated animals during *M. tuberculosis* infection.

Interestingly, despite very similar T cell polarisation to the other two vaccine groups, CysVac2/AlumMPLA_low_ was not protective in the lungs (Figure 2). Lower cytokine expression of CD4^+^ T cells in the lungs of CysVac2/AlumMPLA_low_-immunised mice was observed, but the magnitude of cytokine responses does not necessarily correlate with protection against *M. tuberculosis* infection [48,49]. Notably, IL-2 and TNF expression by CD4^+^ T cells in the lungs of CysVac2/AlumMPLA_low_-immunised mice was not increased compared to PBS control mice. IL-2-expressing CD4^+^ T cells have been proposed as crucial effector cells in the pulmonary anti-*M. tuberculosis* response [48,50]. It would be of interest in future studies to further dissect the differences between the low- and high-dose CysVac2/AlumMPLA vaccine groups that may shed light on the immune requirements for protection.

T cells are essential for the control of mycobacterial infection, and in line with this, a prominent CD4^+^ T cell signature was observed in the vaccinated mice compared to PBS control mice (Figure 6). Despite their importance in controlling mycobacterial infections, IFN-γ or TNF expression by lung CD4^+^ T cells after challenge did not correlate with protective efficacy (Figure 7). Antigen-specific CD4^+^ T cells in the circulation also did not correlate with reduced bacterial burden; Darrah et al. also observed that pulmonary immune readouts in rhesus macaques were more predictive of protection than those of the blood [14]. In contrast, CD4^+^ T cells expressing IL-17A, IL-2 or RorγT or co-expressing IL-17A, IL-2 and TNF in the lungs all significantly correlated with reduced bacterial burden and circulating CD4^+^ T cells expressing IL-17A showed a trending correlation. The presence of B cells and CD8^+^ cells expressing RorγT in the lungs also significantly correlated with reduced bacterial burden, perhaps reflective of the initiation of inducible bronchus-associated lymphoid tissue formation and possible recruitment of Tc17 or MAIT cells, shown in previous studies to be activated after intrapulmonary or IV immunisation [11,13,51,52]. Recent rhesus macaque studies demonstrated IV administration of BCG led to sterilising protection in some animals [13,14]. Analysis of these studies showed that in protected animals, there were increased numbers of antigen-specific T cells in the broncho-alveolar lavage. In particular, the study of Dijkman et al. showed that lung polyfunctional Th17 cells, IL-10 and IgA correlated with protective efficacy after pulmonary BCG immunisation in NHPs [30].

Altogether, these data demonstrate the importance of generating immune memory at the site of infection and support the progression of intrapulmonary TB vaccine candidates. Despite using adjuvants with diverse proposed mechanisms of action, an almost identical circulating immune phenotype following IT vaccination and a highly similar response in the lungs was observed after *M. tuberculosis* challenge. This phenotype was highly divergent from that of unvaccinated mice, and correlations analysis corroborated a CoP for TB: the presence of pulmonary polyfunctional antigen-specific CD4^+^ T cells expressing IL-17A. In total, this study supports the progression of pulmonary subunit vaccines, particularly those adjuvanted with Advax or alum/MPLA, to clinical trials.

## 5. Conclusions

In conclusion, these data demonstrate the efficacy of IT vaccination with CysVac2 vaccines adjuvanted with Advax or alum/MPLA_high_ in a mouse model of *M. tuberculosis* infection. In addition, this study supports the characterisation of antigen-specific Th17 cells in the lungs as a potential novel CoP for *M. tuberculosis* infection. Finally, this study highlights the importance of measuring mucosal immune responses to vaccination in the pursuit of novel CoP against *M. tuberculosis.*

## Figures and Tables

**Figure 1 vaccines-12-00128-f001:**
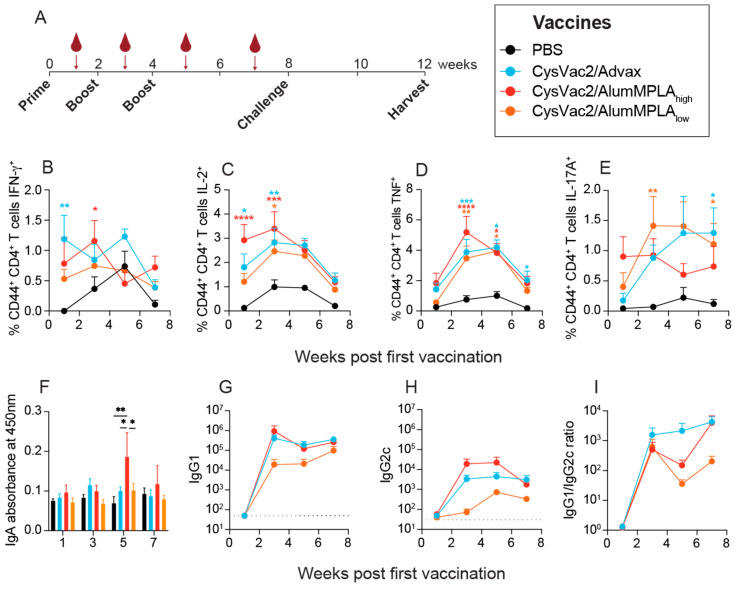
Intratracheal immunisation of CysVac2 with Advax or alumMPLA generates similar kinetics of circulating adaptive immune responses. C57BL/6 mice were immunised intratracheally with PBS (black), CysVac2/low-dose alumMPLA (orange), CysVac2/high-dose alum/MPLA (red) or CysVac2/Advax (blue) three times two weeks apart, and blood was collected as per the schedule outlined in (**A**). After each blood collection (indicated by the maroon blood symbol and arrow), PBMCs were restimulated with CysVac2 protein overnight in the presence of protein transport inhibitor cocktail and then stained for intracellular cytokine production (**B**–**E**). Plasma was also collected for antibody analysis; IgA, IgG1 and IgG2c antibodies specific for CysVac2 protein were measured with ELISAs (**F**–**H**). The ratio of IgG1 to IgG2c in paired samples was also calculated (**I**). Dashed line indicates limit of detection for (**G**–**I**). Graphs depict the mean +/− SEM of pooled data from two independent experiments with 5–6 mice per group. Statistical differences from PBS controls were compared using a 2-way ANOVA with multiple comparisons, corrected using the Dunnett post-hoc test, *p* < 0.05 (*), *p* < 0.005 (**), *p* < 0.0005 (***), *p* < 0.0001 (****).

**Figure 2 vaccines-12-00128-f002:**
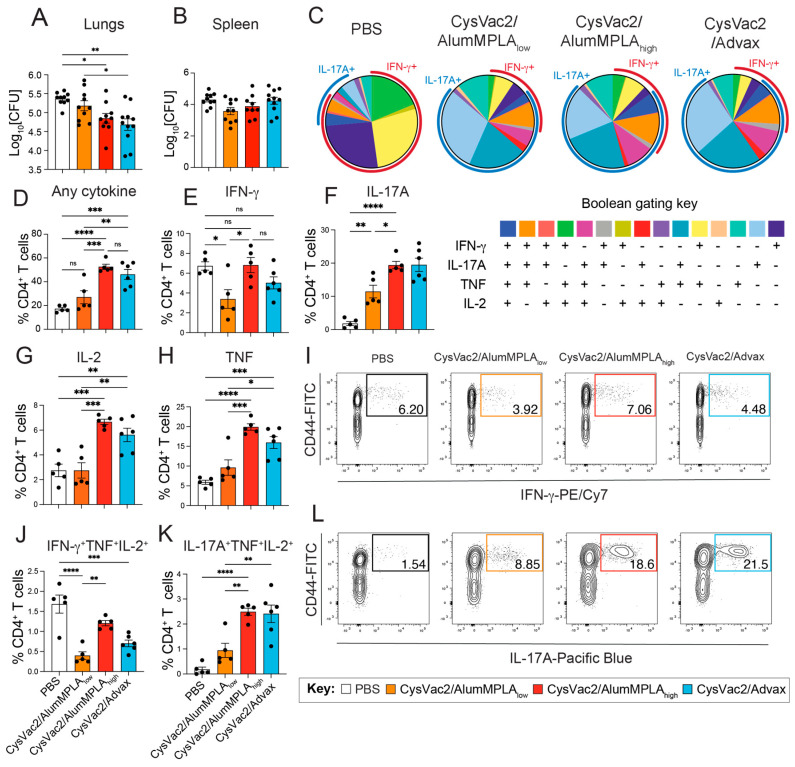
Intratracheal immunisation with CysVac2 vaccines promotes the presence of multifunctional IL-17A-producing CD4^+^ T cells in the lungs of mice infected with *M. tuberculosis.* C57BL/6 mice were immunised with PBS (white), CysVac2/low-dose alumMPLA (orange), CysVac2/high-dose alumMPLA (red) or CysVac2/Advax (blue) and challenged with *M. tuberculosis* H37rV, as described in Figure 1. Four weeks after challenge, lung single-cell suspensions were restimulated overnight with CysVac2 protein in the presence of protein transport inhibitor cocktail and then stained intracellularly for expression of IFN-γ, IL-17A, TNF and IL-2. Concurrent expression of cytokines (**A**) was determined via Boolean gating as per the gating strategy defined in Appendix A. The proportion of total lung CD4^+^ T cells expressing any of the cytokines measured is shown in (**B**). The proportion of lung CD4^+^ T cells expressing IFN-γ, IL-17A, IL-2 or TNF (alone or in combination with other cytokines) is shown in (**C**). The frequency of CD4^+^ T cells expressing any of the above cytokines is shown in (**D**), and individual cytokines are shown in (**E**–**H**). The frequency of multifunctional cytokine-producing CD4^+^ T cells expressing IFN-γ, TNF and IL-2 concurrently is shown in (**J**), and cells expressing IL-17A, TNF and IL-2 concurrently are shown in (**K**). Representative FACS plots of gating strategies for lung CD4^+^ T cell cytokine expression are shown in (**I**,**L**). Graphs are representative of two independent experiments, showing the mean +/− SEM of 5–6 mice per group. Statistical differences were compared using a 1-way ANOVA with multiple comparisons and the Tukey post-hoc test, *p* < 0.05 (*), *p* < 0.005 (**), *p* < 0.0005 (***), *p* < 0.0001 (****), ns refers to not significant differences.

**Figure 3 vaccines-12-00128-f003:**
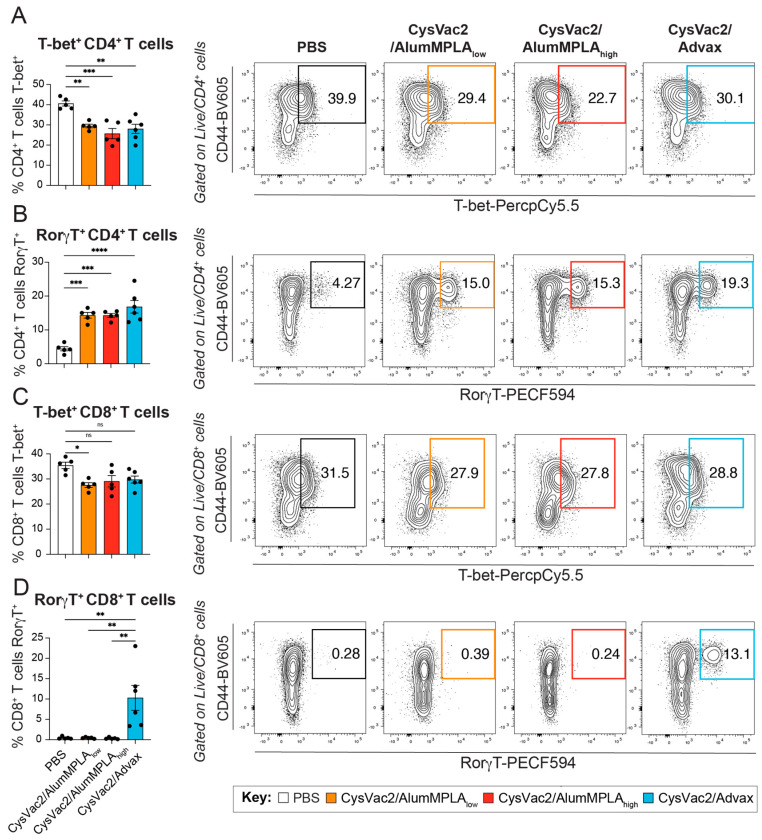
Intratracheal immunisation with CysVac2 vaccines induces significant RorγT expression in lung lymphocytes during *M. tuberculosis* infection. C57BL/6 mice were immunised with PBS (white), CysVac2/low-dose alum/MPLA (orange), CysVac2/high-dose alumMPLA (red) or CysVac2/Advax (blue) and challenged with *M. tuberculosis* H37Rv, as described in Figure 1. Four weeks after challenge, lungs were collected for flow cytometric analysis. Lung CD4^+^ T cells and CD8^+^ T cells were stained intracellularly for expression of RorγT (**A**,**C**) and T-bet (**B**,**D**) transcription factors, respectively. Graphs are representative of two independent experiments, depicting the mean +/− SEM of 5–6 mice per group. Statistical differences were compared using a 1-way ANOVA with multiple comparisons and the Tukey post-hoc test, *p* < 0.05 (*), *p* < 0.005 (**), *p* < 0.0005 (***), *p* < 0.0001 (****), ns refers to not significant differences. Data are shown as means +/− SEM.

**Figure 4 vaccines-12-00128-f004:**
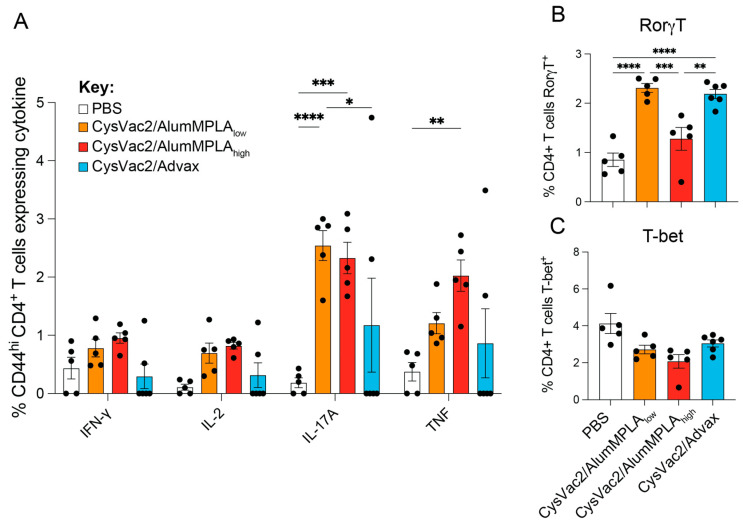
Intratracheal immunisation with CysVac2 vaccines generates Th17 cells in the lung-draining lymph node. C57BL/6 mice were immunised with PBS (white), CysVac2/low-dose alum/MPLA (orange), CysVac2/high-dose alumMPLA (red) or CysVac2/Advax (blue) and challenged with *M. tuberculosis* H37rV, as described in Figure 1. Four weeks after challenge, lungs and mediastinal lymph nodes (mLN) were collected for flow cytometric analysis. mLN cell suspensions were restimulated overnight with CysVac2 protein in the presence of protein transport inhibitor cocktail and then stained intracellularly for cytokine expression. The proportion of mLN CD4^+^ T cells expressing IFN-γ, IL-17A, IL-2 or TNF (alone or in combination with other cytokines) is shown in (**A**). mLN single-cell suspensions were also stained for transcription factor expression, with CD4^+^ T cell expression of RorγT and T-bet shown in (**B**,**C**). Graphs are representative of two independent experiments, showing the mean +/− SEM of 5–6 mice per group. Statistical differences were compared using a 2-way or 1-way ANOVA with multiple comparisons and the Tukey post-hoc test, *p* < 0.05 (*), *p* < 0.005 (**), *p* < 0.0005 (***), *p* < 0.0001 (****).

**Figure 5 vaccines-12-00128-f005:**
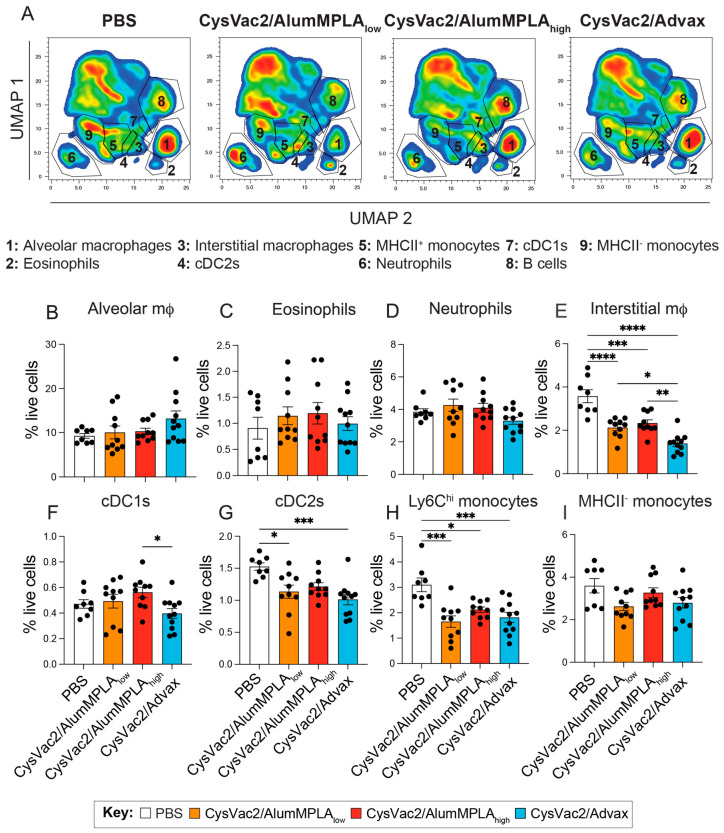
Intratracheal CysVac2 vaccines reduce monocytic infiltration to the lungs 4 weeks post-aerosol *M. tuberculosis* challenge. C57BL/6 mice were immunised with PBS (white), CysVac2/low-dose alum/MPLA (orange), CysVac2/high-dose alumMPLA (red) or CysVac2/Advax (blue) and challenged with *M. tuberculosis* H37rV, as described in Figure 1. Four weeks after challenge, lungs were collected for flow cytometric analysis. Uniform manifold approximation and projection for dimension reduction (UMAP) analysis was performed on pooled flow cytometric samples from one experiment with 5–6 mice per group. Representative samples from each group are shown in (**A**). UMAP gating was confirmed using manual gating, shown in Appendix A. Shown are the frequency of myeloid cell subsets alveolar macrophages (**B**), eosinophils (**C**), neutrophils (**D**), interstitial macrophages (**E**), cDC1s (**F**), cDC2s (**G**), Ly6C high monocytes (**H**) and MHCII^−^ monocytes (**I**). Data shown are pooled from two independent experiments, depicting the mean +/− SEM of 5–6 mice per group. Statistical differences were compared using a 1-way ANOVA with multiple comparisons and the Tukey post-hoc test, *p* < 0.05 (*), *p* < 0.005 (**), *p* < 0.0005 (***), *p* < 0.0001 (****).

**Figure 6 vaccines-12-00128-f006:**
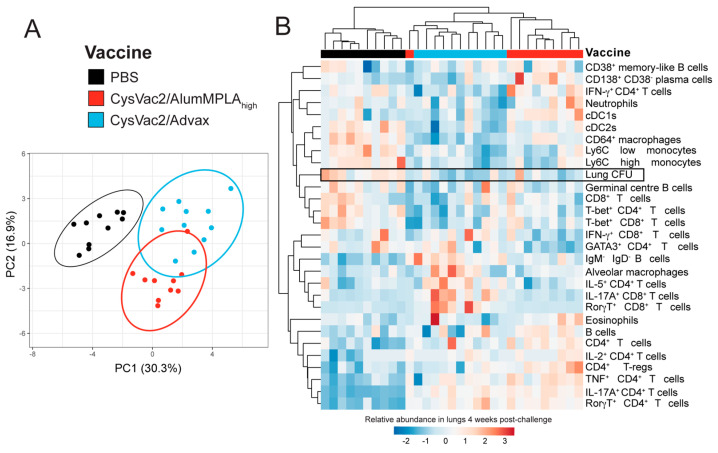
CysVac2 vaccines adjuvanted with Advax or alumMPLA induce different immunological profiles after aerosol *M. tuberculosis* challenge, with a shared protective signature. C57BL/6 mice were immunised with PBS (black), CysVac2/high-dose alumMPLA (red) or CysVac2/Advax (blue) and challenged as described in Figure 1, and the pulmonary immune response after challenge was characterised using flow cytometry. Flow cytometry data from two experiments were normalised and then analysed using the web tool ClustVis to create a principal component analysis plot (**A**) and a heatmap of the various parameters measured (**B**). Data shown are pooled from 2 independent experiments each with *n* = 5–6 mice per group.

**Figure 7 vaccines-12-00128-f007:**
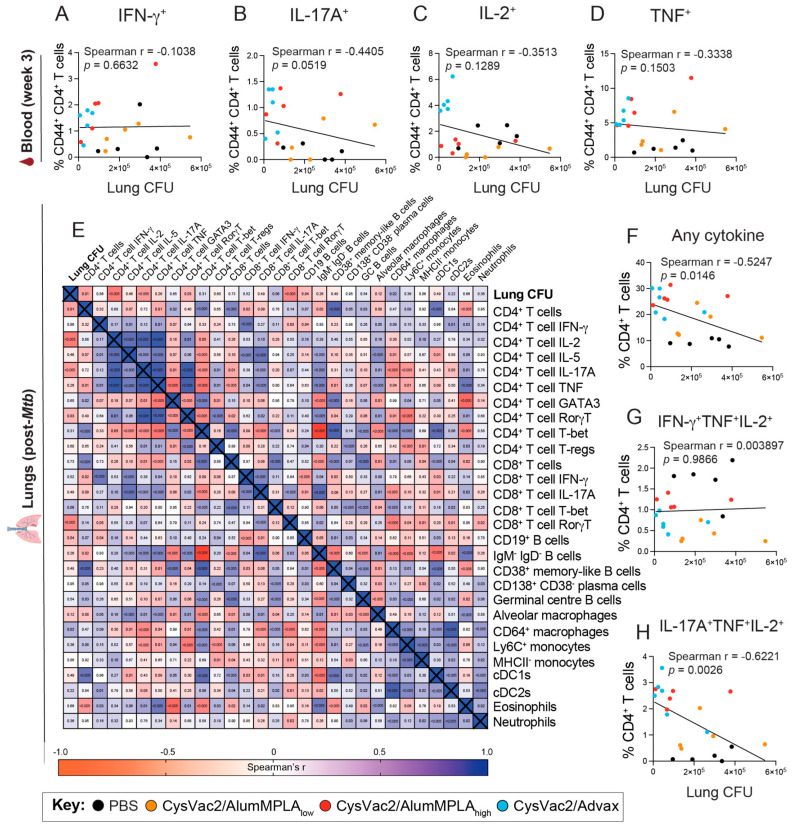
Pulmonary CD4^+^ T cells expressing IL-17A correlate with protective efficacy of intratracheally administered vaccines. C57BL/6 mice were immunised with PBS (black), CysVac2/low-dose alumMPLA (orange), CysVac2/high-dose alumMPLA (red) or CysVac2/Advax (blue) and challenged as described in Figure 1, and the peripheral and pulmonary immune response after vaccination or challenge, respectively, was characterised using flow cytometry. The immunological data from individual mice were paired with the lung bacterial load from 4 weeks post-challenge. Lung bacterial load was correlated with the peak pre-challenge PBMC CD4^+^ T cell cytokine response taken from week 3 of the experiment (one week after the first boost) for cytokines IFN-γ, IL-17A, IL-2 and TNF (**A**–**D**). Black dots represent PBS-immunised mice, blue CysVac2/Advax-, orange CysVac2/AlumMPLA_low_-, and red CysVac2/AlumMPLA_high_-immunised mice. Correlation analysis was also performed for lung immune responses measured 4 weeks after aerosol challenge with *M. tuberculosis* (**E**–**H**). A Spearman’s correlation test was performed using GraphPad Prism Software to determine correlation and *p* values, where *p* < 0.05 were considered significant. For individual plots, data are representative of 2 independent experiments with *n* = 5–6 mice per group, and for (**E**), data are pooled from 2 independent experiments each with *n* = 5–6 mice per group.

## Data Availability

The data that support the findings of this study are available from the corresponding author upon reasonable request.

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
