# Peer review of "Lung IL-17A-Producing CD4+ T Cells Correlate with Protection after Intrapulmonary Vaccination with Differentially Adjuvanted Tuberculosis Vaccines"

_vaccines, 2024, doi:10.3390/vaccines12020128_

Round 1

Reviewer 1 Report (Previous Reviewer 1)

Comments and Suggestions for Authors

I have no concerns with this manuscript.

Author Response

We thank the reviewers for their comments. 

Reviewer 2 Report (Previous Reviewer 2)

Comments and Suggestions for Authors

This manuscript describes T cell responses to CysVac2 immunization to define CoP. This study has compared vaccine efficacy among CysVac2 vaccines using three adjuvant conditions: Advax, and Alum+MPLA of high and low concentration, which were vaccinated via intratracheal instillation, and revealed that pulmonary polyfunctional Th17-expressing cells(IL-17+TNF+IL-2+CD4+ T cells) are crucial for protection, but IFN-r+TNF+IL-2+ CD4+T cells were not correlated to the bacterial load of lung. These data provide important information in the TB vaccine research field and are an interesting topic that needs the attention of the scientific community, but analysis of immune response in the tissue before the challenge is needed in terms of identification of CoP. 

When comparing to your previously published papers about CysVac2, (1) Factors correlated with protection is the frequency of pulmonary IL-17+TNF+IL-2+CD4+ T cells but not IFN-r+TNF+IL-2+CD4+T cells, which is specific for intratracheal immunization? How about CysVacA immunization via s.c. or i.m.? Please discuss this issue in the DISCUSSION section. (2) In the previous papers [PMID: 33298977, 33542494], CysVac2 was administered at a dose of 3 μg. Could you please explain the rationale behind increasing the dosage to 10 μg in this study? Immunization of 3 ug CysVac/Advax (i.t.) in the previous paper (PMID: 33298977) was protective in the lung and spleen, but this manuscript, 10 ug CysVac/Advax (i.t.) was not protective in the spleen.   

In minor comments: 1) please indicate peripheral blood taken site from the mice. Can it obtain enough cells to conduct the assay of Fig.1. 2) In Figure 3, please indicate the group in the bar chart.

Author Response

We thank the reviewers for their comments. Changes based on your comments have been made as shown below.

  1. Factors correlated with protection is the frequency of pulmonary IL-17+TNF+IL-2+CD4+ T cells but not IFN-r+TNF+IL-2+CD4+T cells, which is specific for intratracheal immunization? How about CysVacA immunization via s.c. or i.m.? Please discuss this issue in the DISCUSSION section.

Author response: Discussion of this point has been added on page 15:

“Route of delivery appears to be a major contributing factor to the observed immune signature induced by CysVac2/Advax and other TB vaccines administered mucosally [25, 32, 37, 39]. Previously we have shown that intramuscular delivery of CysVac2/Advax generates multifunctional CD4+ T cells expressing IFN-g, TNF and IL-2 but IT administration instead promotes a lung-local Th17 signature [11, 19]. It would be of interest to examine if the immune signature identified here correlates with protective efficacy after parenteral vaccination, or if it is specific to pulmonary vaccine models.”

  1. In the previous papers [PMID: 33298977, 33542494], CysVac2 was administered at a dose of 3 μg. Could you please explain the rationale behind increasing the dosage to 10 μg in this study? Immunization of 3 ug CysVac/Advax (i.t.) in the previous paper (PMID: 33298977) was protective in the lung and spleen, but this manuscript, 10 ug CysVac/Advax (i.t.) was not protective in the spleen.

Author response: We previously performed a titration study for IT dosage of both Advax adjuvant and CysVac2 protein and found that 1 mg Advax with 10 µg CysVac2 provided greater protection in the lungs than the 3 µg dose. We typically observe large variability in protection in the spleen using TB vaccines, and thus we consider that protection in the lung is a more reliable readout of vaccine efficacy.

  1. Please indicate peripheral blood taken site from the mice. Can it obtain enough cells to conduct the assay of Fig.1. 

Author response: Peripheral blood was collected from the tail vein, and the PBMCs from these samples were sufficient to perform the restimulation experiments shown in Figure 1. This has now been referred to in the materials and methods on page 3.

  1. In Figure 3, please indicate the group in the bar chart.

Author response: Thank you, we have added the groups to the bar chart.

Reviewer 3 Report (New Reviewer)

Comments and Suggestions for Authors

The main drawback of the article under review is the lack of Suppl. It seems difficult to review an article without suppl. One of the corrected versions of the manuscript is attached instead of the Suppl.

Minor:

1.       The caption to the figure must indicate what the red blob represents (Figure 1A).

2.       In Figures 2-5 you should leave out either dots or bars. Dots and bars together make the figure harder to understand.

3.       In Figures 3-5 you must indicate what the different coloured columns and gates represent.

4.       Fig 2C - What kind of CD4 T cells are these? What's their phenotype?

5.       BCG, the currently used tuberculosis vaccine, induces significant levels of IFN-γ-expressing CD4+ T cells when administered subcutaneously or intradermal. intradermally - what was meant, probably a typo. The live BCG vaccine is not administered intradermally.

Author Response

We thank the reviewers for their comments. Changes based on your comments have been made as shown below.

  1. The main drawback of the article under review is the lack of Suppl. It seems difficult to review an article without suppl. One of the corrected versions of the manuscript is attached instead of the Suppl.

Author response: Apologies for this oversight, we have now attached the supplementary information.

  1. The caption to the figure must indicate what the red blob represents (Figure 1A).

Author response: We have now added to the figure legend a reference to the red blob indicating tail bleeds.

  1. In Figures 2-5 you should leave out either dots or bars. Dots and bars together make the figure harder to understand.

Author response: Thank you for your suggestion. We prefer to display the data as dots and bars, since the dots show the individual biological replicates and provide a clear depiction of the variability of the data, while the bars provide the reader with a clear indication of the mean of each experimental group.

  1.  In Figures 3-5 you must indicate what the different coloured columns and gates represent.

Author response: We have added to Figure 3 labels for the bars, and in Figure 4 an additional legend. In Figure 5 the bars are already labelled.

  1. Fig 2C - What kind of CD4 T cells are these? What's their phenotype?

Author response: The CD4 T cells shown in Figure 2 are total lung CD4 T cells, gated as per Supplementary Figure 1. This has now been referred to in the figure legend.

  1. BCG, the currently used tuberculosis vaccine, induces significant levels of IFN-γ-expressing CD4+ T cells when administered subcutaneously or intradermal. intradermally - what was meant, probably a typo. The live BCG vaccine is not administered intradermally.

Author response:

BCG is administered intradermally in humans but is sometimes administered subcutaneously in murine models. To clarify this statement, we have removed reference to subcutaneous injection.

Round 2

Reviewer 2 Report (Previous Reviewer 2)

Comments and Suggestions for Authors

.

Author Response

no comments 

Reviewer 3 Report (New Reviewer)

Comments and Suggestions for Authors

It is unclear why the authors do not listen to the reviewer's comments.

The figures are not yet captioned to explain what the different colors mean. Each Figure should be labeled. It would be easier to add a text to the legend to each figure explaining what each color represents. This would make it much easier to understand the drawing.

You must select points or bars in the figures. In the figure, you show error bars - this is what provides a clear representation of the variability of the data. Points, bars, error bars and significance together make the figure a “hedgehog”.

Author Response

The figures are not yet captioned to explain what the different colors mean. Each Figure should be labeled. It would be easier to add a text to the legend to each figure explaining what each color represents. This would make it much easier to understand the drawing.

Author response:

We thank the reviewer for their comments. We have made changes based on  feedback that we believe make the figures much clearer and easier to understand.

We have added to each of the figure legends indications of the colour scheme in the figures and the experimental groups they refer to. i.e. “C57BL/6 mice were immunised with PBS (white), CysVac2/low-dose Alum/MPLA (orange), CysVac2/high-dose Alum/MPLA (red), or CysVac2/Advax (blue)”.

In addition, we have added a colour key indicating the experimental groups in figures 2, 3, 4, 5 and 7.

You must select points or bars in the figures. In the figure, you show error bars - this is what provides a clear representation of the variability of the data. Points, bars, error bars and significance together make the figure a “hedgehog”.

We thank the reviewer for their comments. In this instance we would like to maintain presentation of bars and the individual data points as we would like to present the data to the reader as transparently and with as much information as possible.

This manuscript is a resubmission of an earlier submission. The following is a list of the peer review reports and author responses from that submission.

Round 1

Reviewer 1 Report

Comments and Suggestions for Authors

In the manuscript “Lung IL-17A-producing CD4+ T cells correlate with protection after intrapulmonary vaccination with differentially adjuvanted tuberculosis vaccines." by Erica L. Stewart and colleagues, the authors evaluated the immunogenicity and efficacy of the TB vaccine candidate CysVac2 combined with either Advax adjuvant or a mixture of alum plus MPLA, administered intratracheally into mice. Peripheral immune responses were tracked along the challenge experiments and lung and spleen immune responses were measured after challenge. The efficacy shows a significant reduction of Mtb CFU in the lung but not in the spleen, using CysVac2 combined with either Advax. Also, the authors identify CD4+ T cells expressing IL-17A and RorgT (Th17 cells) as the correlate of protection for this particular vaccine candidate. Even though these results are encouraging, the experiments lack an important control, which is mice vaccinated with BCG, the current gold-standard for vaccine immunogenicity and efficacy studies. Currently, there is not better vaccine than BCG at inducing some protection. Thus, the goal in the field is to develop a new vaccine able to improve or replace BCG. This study lacks the important BCG control, which relevant. It is clear that the study was very well conducted and executed, and the data is strong, but the lack of the BCG control group removes relevance and impact of the study. Additionally, this vaccine could have been tested as a booster of BCG, and the results would have been very interesting.

The manuscript well written and the references are up to date.

Concerns and comments:

1) Should the authors have produced immunogenicity and efficacy data of the BCG control group, please include it in the manuscript. If not, please explain clearly in the manuscript why BCG was not used as a control in this study.

2) Please present the references consistently throughout the text.

3) Figure 6, panel A and B, the legend for PBS is a white square, but the control samples are marked in black color. Please correct this.

4) Also, on the figure 6, the authors describe the data analysis for the flow cytometry data but do not refer to the data analysis of the Lung CFU data. Please include a short description of the color code used on the Lung CFU data.

Reviewer 2 Report

Comments and Suggestions for Authors

This manuscript submitted by Stewart and collaborators outlines the study on analyzed T cell responses to CysVac2 immunization to define CoP. This study has compared vaccine efficacy among CysVac2 vaccines using three adjuvant conditions: Advax, and Alum+MPLA of high and low concentration, which were vaccinated via intratracheal instillation, and revealed that Th17 cells, particularly those expressing IL-17A, are crucial for protection. Furthermore, vaccines inducing these specific T cells in the lungs were most effective, highlighting the importance of local immune responses. Adjuvants like Advax or Alum/MPLA of high concentration promoted similar immune reactions, advocating for the advancement of pulmonary subunit vaccines, and emphasizing the need for site-specific immune memory in TB vaccine development. This manuscript covers an interesting topic that needs the attention of the scientific community. 

However, my major concern is whether IL-17 increase as a maker for CoP is acceptable to other TB vaccines, or specific to CysVac2 vaccine and/or IT administration. Why don’t you include BCG as a positive control? Because as you know, it has been demonstrated that vaccine-induced protection against Mtb infection specifically involves antigen-specific multifunctional IFN-γ+IL-2+TNF-α+ and IL-2+TNF-α+ CD4+ T cells in the lungs, and also IFN-γ/IL-17-Co-producing CD4+ T-cells are the determinants for protective efficacy of tuberculosis subunit vaccine.

In minor comments:

1. It would be more convenient for readers if you tell us what CysVac2 is in the Intro or M&M section. (Ex. CysVac2 is a fusion protein composed of Ag85B and CysD.)

2. Please check Figure 2 mentioned in the Result or Legend section. The description and figure do not seem to match. 

The majority of CD4+ T cells producing IL-17A expressed it alone or in combination with TNF, however, there was also a subset of CD4+ T cells expressing IL-17A alongside TNF and IL-2 (Figure 2G, H, J)--> (Figure 2C, F, K or Figure 2F, K)?

3. In the previous papers [PMID: 33298977, 33542494], CysVac2 was administered at a dose of 3 μg. Could you please explain the rationale behind increasing the dosage to 10 μg in this study?

4. Additionally, the previous paper mentioned an increase in macrophages due to elevated IL-17 levels [PMID: 33298977]. Is the decrease of macrophages observed in Figure 5 a result of the high-dose antigen administration?
